# CM-NET: Cross-Modal Learning Network for CSI-Based Indoor People Counting in Internet of Things

**Jing Guo** [†], **Xiaokang Gu** [†], **Zhengqi Liu, Minghao Ji** , **Jingwen Wang, Xiaoyan Yin and Pengfei Xu** *

School of Information Science and Technology, Northwest University, Xi'an 710000, China
* Correspondence: fpxu@nwu.edu.cn
† These authors contributed equally to this work.

**Abstract:** In recent years, multiple IoT solutions have used computational intelligence technologies to identify people and count them. WIFI Channel State Information (CSI) has recently been applied to counting people with multiple benefits, such as being cost-effective, easily accessible, free of privacy concerns, etc. However, most current CSI-based work is limited to human location-fixed environments since human location-random environments are more complicated. Aiming to fix the problem of counting people in human location-random environments, we propose a solution using deep learning CM-NET, an end-to-end cross-modal learning network. Since it is difficult to count people with CSI straightforwardly, CM-NET approaches this problem using deep learning, utilizing a multi-layer transformer model to automatically extract the correlations between channels and the number of people. Owing to the complexity of human location-random environments, the transformer model cannot extract characteristics describing the number of people. To enhance the feature learning capability of the transformer model, CM-NET takes the feature knowledge learned by the image-based people counting model to supervise the learning process. In particular, CM-NET works with CSI alone during the testing phase without any image information, and ultimately achieves sound results with an average accuracy of 86%. Meanwhile, the superiority of CM-NET has been verified by comparison with the latest available related methods.

**Keywords:** people counting; CSI; knowledge distillation; cross-modal learning network; computational intelligence



## 1. Introduction

People counting provides key information for a wide range of services and applications, such as crowd control for places of interest and marketing research for malls. However, human behavior can be unpredictable; thus, people counting may encounter various challenges, such as object occlusions, pedestrian overlaps, and demands for real-time processing. Traditional solutions to these issues fall roughly into the following categories: environmental sensor-based, vision-based, and wireless signal-based methods.

Along with the advances in sensing technology, many sensor-based networks provide rather good accuracy in estimating the number of people by analyzing variations in the surroundings, such as temperature [1], sound [2], and carbon dioxide [3]. However, the feasibility of sensor-based counting methods is hindered by constraints such as expensive equipment, the complexity of the operation, and limited scope. Vision-based methods have been widely used in many public places [4,5], yet these methods are inherently flawed. First, cameras work only in a line-of-sight pattern, leaving many areas blind to monitoring. Second, smoke or a lack of light in the environment will severely degrade the image quality. Thirdly, overlapping objects further hamper the model's performance. Wireless signals based methods perform people counting based on radio-frequency signals (WIFI [6–9], UWB radar [10,11], etc.). The advantage of these methods is that they are not affected by light, do not violate human privacy and can achieve good recognition results.

For indoor scenarios, the WIFI-based methods have natural advantages: (1) No additional devices are required. With the widespread deployment and coverage of WIFI in public places or home, the existing infrastructures provide the data base for the WiFi-based methods. (2) The monitored person is not required to carry any smart devices or sensors. The principle of the WiFi-based methods is the WIFI signal will reflect, diffract and refract when encountering obstacles or moving individuals in the process of propagation, to form a regular signal change pattern. In addition, the number of people can be identified by analyzing the signal change pattern.

The most commonly used WIFI-based methods are the coarse-grained Received Signal Strength (RSS) and the fine-grained Channel State Information (CSI). For the former, while RSS data can be easily obtained for most off-the-shelf wireless devices, RSS measurements are affected by multipath fading and environmental noises. The RSS dataset is often expanded to improve the count accuracy, which inevitably incurs additional labor and time costs. A recently proposed solution [9] to the above problem estimates the number of people using CSI, which can provide richer information than RSS, including each subcarriers amplitude and phase information. Meanwhile, CSI is more sensitive to the number of people in the environment than RSS. Present-day works, however, assume all human objects are fixed, which is not the case in practice. Therefore, it cannot be applied to scenarios involving random changes in human positions.

A technical challenge that needs to be solved to turn the idea into a working system, i.e., how to extract good features characterizing the number of people in the conditions of different locations. Inspired by knowledge distillation [12], we guide the CSI-based network by other modal networks acting as teachers so that the CSI-based network can extract useful features from the crowd. Several existing methods of counting people have been compared and analyzed, including those using visible images, infrared images, and other environmental sensing information. It has been found that image-based methods are easy to implement and can achieve high accuracy under normal conditions. Therefore, we propose a cross-modal learning network CM-NET, which uses the image-based network as a teacher network to guide the learning of the student network (CSI-based network). As a result of CM-NET, CSI-based networks are capable of achieving more accurate results and alleviating degradation in accuracy caused by the change in human locations.

In CM-NET, both the teacher network and the student network require training. The teacher network is first pre-trained on the COCO dataset and then fine-tuned on our data. The student network is trained using the feature knowledge learned from the trained teacher network. CM-NET avoids privacy issues by using only image information in training and only CSI data in testing.

The main contributions of this paper are as follows:

1. For the first time, we use multimodal knowledge distillation to perform CSI-based people counting. We use multimodal knowledge to compensate for the limits of unimodal network that need a lot of training data and weak feature representation, providing a new idea for CSI-based people counting methods.

2. We propose a cross-modal learning network, CM-NET, which uses the feature information output by the vision network as a supervisory signal to guide feature learning in CSI-based network, alleviating the performance degradation of CSI-based network due to location changes.

3. To the best of our knowledge, we build the first international dataset of indoor multimodal (video and CSI), which includes multimodal data from 60 different locations and 9 different numbers of people. On this dataset, the CM-NET proposed in this paper achieves an average accuracy of 86%, which is better than other current methods for counting people.

The rest of the paper is structured as follows. Section 2 provides an overview and analysis of existing methods and networks related to people counting. Section 3 describes our proposed people counting method more detailed. Section 4 gives the relevant experimental

results and provides a detailed description and analysis of the results. Finally, Section 5 concludes the paper.

## 2. Related Work

Current methods can be classified into three categories: Environmental sensor-based methods, Vision-based methods and Wireless signal-based methods. In this section, we will discuss the existing works on the above three types of people counting methods.

**(1) Environmental sensor-based methods.** These methods can be broadly classified into the following three categories based on the type of sensor: passive infrared sensor-based, sound sensor-based, and carbon dioxide sensor-based.

Passive infrared sensors infer the number of moving bodies based on the ambient temperature change caused by moving objects in the sensor area. F. Wahl et al. [1] arranged passive infrared sensors at each pedestrian walkway and then designed a probabilistic distance-based algorithm to estimate the number of people in the environment. The test results show that the probabilistic distance-based algorithm can compensate for the error due to the infrared mask effect. The statistical error increases significantly when multiple people simultaneously pass through the pedestrian passage. Statistical errors can also be caused by interference from sunlight and heat.

The sound sensor-based people counting method sends an ultrasonic signal to the monitoring area using a sound sensor. It reflects when the ultrasonic signal encounters a moving object, such as a person's body. When the sound sensor receives the reverberation of the transmitted signal, the number of people in the environment can be inferred based on characteristics such as reception time or signal attenuation. O. Shih et al. [2] designed a system to count people in an area using changes in the room's acoustic properties. Experiments have shown that the acoustic sensor-based solution performs better in smaller indoor spaces and crowds. However, the system's scalability is somewhat limited, as the accuracy decreases significantly with increasing space and occupancy. In addition, the number of people estimates may be distorted due to the presence of a large amount of sound-absorbing materials often present in indoor environments.

Since people's respiration produces carbon dioxide in a room, it is possible to infer the number of people by the concentration of carbon dioxide. Such scheme uses sensors to measure the concentration of carbon dioxide in a room and estimate the number of people. H. Rahman et al. [3] proposed a people counting system based on indoor carbon dioxide emissions. Experiments show that the carbon dioxide sensor-based people counting method is slow to respond to dynamic changes as it takes a certain amount of time for the carbon dioxide concentration to change if someone enters or leaves the room.

**(2) Vision-based people counting methods.** N. Dalal et al. [4] proposed a technique for intact human body-based detection. This technique extracts directional gradient histograms from pedestrian sample images as features for person counting using a linear support vector machine [5] for classification. However, intact body-based detection techniques are only suitable for cases where bodies do not overlap, so the system is vulnerable to occlusions or complex backgrounds, resulting in poor recognition performance. Wu Bo et al. [13] proposed a partial body-based detection technique, which uses part of the pedestrian's body structure to perform detection. For example, the Adaboost [14] network can be trained on the head and shoulders of pedestrians. Then, the number of pedestrians' heads and shoulders are detected in the image to calculate the number of people. Although this technique can reduce statistical errors caused by multi-person overlap, it still poses problems determining image region division and sliding window size.

The regression network-based people counting technique [15] using a proven regression network counts the number of people in an image by considering the crowd in the image as a whole object. The technique extracts foreground image features from the collected images [15], then trains a regression model using the obtained feature set, and then uses the built regression model to obtain the headcount information from the test sample. The features used can be classified as foreground image features, edge features [16],

gradient features [17], and texture features [18]; regression strategies comprise Gaussian process regression [19], nonlinear regression, and neural networks; images can be processed according to partitioning, sliding windows, global estimation, etc. Unlike detection-based algorithms, regression networks have good generalization capabilities, excellent portability, and endless possibilities. A. Davies et al. [15] found a correlation between the number of people in the environment and the foreground pixel area and chose to use regression information to identify the number of pedestrians in an image. This work was the first to introduce regression methods to crowd analysis and indirectly inspired people counting. Regression-based methods also fail to overcome the impact of overlap occlusion of the human body.

**(3) Wireless signal-based methods.** Wireless signal-based methods use deployed wireless devices to build an environment-aware system for people counting. These techniques not only compensate for the vulnerability of sensor-based people counting techniques to environmental influences but also fill the gap of video-based counting techniques that do not work in privacy and obscuration situations. As far as the current research in this field is concerned, it can be divided into the following three classes based on the characteristics of the chosen radio signal: UWB signal-based people counting methods, RSS-based people counting methods, and CSI-based people counting methods.

People counting based on UWB signals [10] is mainly performed through radar networks. The method first removes static objects and people reflections by the background phase cancellation method [11]. After that, it calculates the number of people based on the waveform characteristics of the received signal. J. W. Choi et al. [20] proposed an IR-UWB radar system that sends a broadband pulsed signal along with receiving the backscattered signal from the environment to infer the number of human targets. Experiments have shown that the system can detect up to three targets, with errors reaching 8%. UWB signal-based counting technology has excellent recognition performance; however, this technology requires expensive and special hardware devices to support it, so it is difficult to prevail in the production life of people.

In 2008, M. Nakatsuka et al. [6] first validated the feasibility of using RSSI for people counting and designed a linear regression-based model. This model relies mainly on the mean change of RSSI measurements to calculate the number of people. Experimental results demonstrated that the RSS between two radio nodes decreases as the number of people bodies located between these nodes increases, and they proposed linear network can detect up to 9 people. In 2015, T. Yoshida et al. [7] implemented an RSS-based people counting technique based on support vector regression. This method can count up to 7 people with an accuracy of up to 77%. Alsamhi et al. [21] proposed extending the ANN's method to UAVs to predict the intensity of signals in different areas of the city, planning for WIFI-based outdoor people counting in the future. Although RSSI-based people counting technology can significantly reduce equipment deployment costs by depending on WiFi infrastructure, RSSI, as a coarse-grained description of the channel, can only reflect the signal's fading after propagation. In a complex and noisy indoor environment, the performance of a system designed by RSS can be greatly compromised.

With the development of technology and the unremitting efforts of researchers, D. Halperin et al. [8] extracted Channel State Information (CSI) on commercial Wi-Fi supporting 802.11n protocol using the self-developed CSI Toolkit. With the successful extraction of CSI, Xi Wei et al. [9] proposed a passive number counting system based on CSI. In this system, the authors utilize the percentage of non-zero elements in the inflated CSI matrix as features to describe the variation of the wireless channel, and use the gray Verhulst model for feature training thereby building a library of feature-number relations, which can be used to identify the number of people at the testing phase. S. D. Domenico et al. [22] proposed a people counting system based on differential CSI measurements. The system uses Euclidean distance to represent the difference between two CSI measurements and uses features such as first- and second-order statistical moments for classification. Experiments show that the system can accommodate up to seven people in an indoor environment with

a classification error of about 15%. Compared with the RSSI-based counting method, the CSI-based method offers more stable performance and relatively higher accuracy. In 2016, Alsamhi et al. [23] proposed a technique for maintaining signal quality on high-altitude mobile platforms, improving the signal quality and mitigating the impact of signal quality on the number of people counted.

## 3. Design and Training of CM-NET

Generally, the performance of visual-based people counting models, once trained, is almost position-independent. However, visual-based people counting models may be susceptible to factors such as illumination and occlusion, leading to significant performance degradation. Furthermore, privacy concerns involving those being watched may arise with vision-based networks. In contrast, CSI-based networks rely on CSI to infer the number of people, which is readily available even in low-light settings. These methods have very few privacy concerns because they do not specifically use the monitored people's images. Nevertheless, monitored individuals' location changes decrease CSI-based networks' performance. The location sensitivity limits of CSI-based people counting jobs may be partially overcome if CSI-based counting networks can achieve comparable anti-interference and discriminating skills to vision-based networks [24,25].

Based on the preceding study and motivated by the knowledge distillation technique, we propose an end-to-end cross-modal learning network called CM-NET that trains the CSI-based network using the class probabilities from the vision-based network as soft labels. CM-NET's architecture consists of teacher and student networks; the training framework is depicted in Figure 1. CM-NET's teacher network performs people counting with visible images.

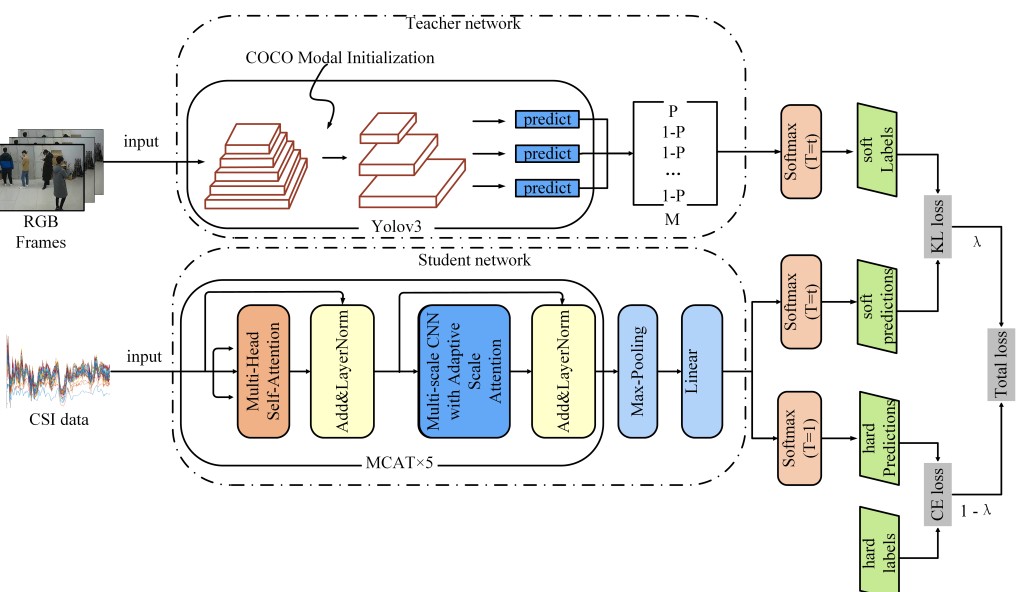

**Figure 1.** The training framework of CM-NET. It has two components: the teacher's training network and the student's training network. The teacher is a vision-based people counting network, whose network is first initialized using the trained weights from the COCO dataset and then fine-tuned with our collected video data. The student is a CSI-based people counting network, whose network training process is supervised by the soft labels from the teacher network.

In CM-NET's teacher network, the Yolov3 target detection module first obtains the probability of each person in the image. Then, the probability matrix of the total number of people category is generated by the uniform distribution. Finally, the probability matrix is fed to the softmax classifier. The output of the softmax classifier is used as a soft label to guide the training of the student network. The student network of CM-NET uses CSI to count people, consisting of the transformer feature extraction module and the softmax classifier

module. It should be noted that CM-NET only introduces the soft labels generated by the teacher network as supervised information during the student network's training process.

Section 3.1 describes the teacher network of CM-NET, and Section 3.2 describes the student network of CM-NET. Finally in Section 3.3, we will describe the design of the CM-NET loss function.

### 3.1. Teacher Network: Vision-Based People Counting Network

CM-NET's teacher network uses Yolov3 as the detection model. We customized the output module of Yolov3 in order to transfer the knowledge gained to the student network. We statistically analyze the degree of confidence of each individual in the Yolov3 photos, denoted as $p_i$, where $i$ represents the $i$th people detected. In addition, the statistics are summed and averaged to obtain the confidence scores for the total number of people denoted as $P$, the corresponding function is shown in Equation (1). Then, the confidence score matrix for total number of people class denoted as $M = [m_1, m_2, m_3, \ldots, m_K]$, was obtained by expanding $P$ through a label smoothing technique, the corresponding function is shown in Equation (2). Finally, $M$ is fed into the softmax layer to obtain the prediction results.

$$P = \sum_i^n p_i / n \tag{1}$$

$$m_j = \begin{cases} P & \text{if } j = n \\ \frac{1-P}{K-1} & \text{if } j \neq n \end{cases} \quad (j = 1, 2, 3, \ldots, K) \cdot \tag{2}$$

where $K$ represents the total classes of crowd. It is important to note that the soft labels generated by the teacher network can more accurately represent the similarity across classes than the actual labels. The knowledge distillation approach adds a temperature coefficient to the original softmax function to soften the class probabilities in order to better represent this similarity between classes. The softened soft label is represented as $O_V = [v_1, v_2, v_3, \ldots, v_K]$, and it is calculated as shown in Equation (3).

$$v_i = \frac{e^{(m_i/T)}}{\sum_{k=1}^K e^{(m_k/T)}} \tag{3}$$

where it can be seen that the distribution of probability output is "softer" under high temperatures. The "softer" probability distribution is more effective in extracting knowledge and training the network.

### 3.2. Student Network: CSI-Based People Counting Network

CSI is 2D time series data denoted as $C \subseteq \mathbb{R}^{d \times n}$, where $d$ and $n$ represents channel number and received package number, separately. The CSI data contains noise and outliers due to the presence of multipath effects and ambient noise in the room, as well as defects in the hardware equipment itself. If the raw data are used directly, the performance of CSI-based people counting networks will be significantly degraded. For this reason, we performed the necessary preprocessing. We first apply Equation (4) to remove the outliers from the raw CSI data and then smooth CSI data using linear interpolation.

$$C = [\mu - \eta \times \delta, \mu + \eta \times \delta] \tag{4}$$

where $\mu - \eta$ is the median of a set of observations, $\delta$ is the absolute deviation from the median, and $\eta$ is the empirical constant, which is taken as 2 in this paper.

Then, we use a soft-threshold denoising method based on wavelet analysis to eliminate the noise. In this paper, we use a Gaussian function as the wavelet basis function for 12-layer denoising of CSI data and a soft thresholding method for high frequency coefficients. The processed CSI data are fed into the student network for training. The student network uses the transformer framework as the base network, which consists of five MCAT layers [26], a

pooling layer and a fully connected layer. The student network feeds the softmax layer with the people in the crowd extracted by the transformer's digit extraction algorithm to obtain the prediction results. It is worth noting that during the training process, the predictions of the student network are divided into hard predictions denoted as $O_S = [s_1, s_2, s_3, \ldots, s_K]$ and soft predictions denoted as $O'_S = [s'_1, s'_2, s'_3, \ldots, s'_K]$. The $O_S$ generated by the student network uses the original softmax. This is expressed in Equation (5):

$$s_i = \frac{e^{z_i}}{\sum_{k=1}^{K} e^{z_k}} \tag{5}$$

where $z_i$ is the output of the fully connected layer of the student network.

Similar to $O_V$ generated by the teacher network, $O'_S$ generated by the student network uses the temperature coefficients of the softmax layer, as shown in Equation (6).

$$s'_i = \frac{e^{(z_i/T)}}{\sum_{k=1}^{K} e^{(z_k/T)}} \tag{6}$$

### 3.3. Design of the Loss Function

The loss function $L$ of CM-NET consists of two components: the KL scatter loss $L_{kl}$ between the output of the student network and the output of the teacher network, and the cross-entropy loss $L_{ce}$ between the output of the student network and the hard label, where the hard label is the true label of the sample noted as $Y$. The KL scatter loss $L_{kl}$ can be formulated as follows:

$$L_{kl} = \sum O_{S'} \log \frac{O_{S'}}{O_V} \tag{7}$$

the cross-entropy loss $L_{ce}$ can be formulated as follows:

$$L_{ce} = \sum O_V \log Y + (1 - O_S) \log(1 - Y) \tag{8}$$

Finally, the loss function $L$ of CM-NET can be formulated as follows:

$$\min_S L = \lambda L_{kl} + (1 - \lambda) L_{ce} \tag{9}$$

where $\lambda$ is the balance factor. In the training process of CM-NET, we fixed the parameters of the teacher network and updated the student network using the gradient descent method. The student and teacher networks are dropout-regularized during the training process to avoid overfitting.

## 4. Experiments

The experimental section begins with Section 4.1, which describes how the equipment is deployed and data collected. In Section 4.2, we describe the preprocessing details of the dataset and the CM-NET hyperparameter settings. Section 4.3 describes the metrics for evaluating CM-NET's performance and counting the number of people. Section 4.3.3 shows comparisons between our experiments and existing work.

### 4.1. Equipment Deployment and Data Collection Process

Our data collection is based on commercially available hardware, including a TP-Link camera, wireless router, and laptop with an Intel 5300 NIC. The TP-link wireless router serves as the transmitter, and the laptop the receiver. The receiver and transmitter were mounted on a 0.75 m stand separated by 5.6 m. In order to make sure that the camera would cover the entire area used for collecting CSI data, we fixed the camera to the wall four meters high.

We set up the equipment mentioned above in our lab to gather CSI and video data. The size of the area where we collected CSI data was 2.4 m × 3.2 m, and the area is shown in Figure 2. We enlisted 9 volunteers for this experiment, including 2 females and 7 males.

We gathered information under two cases: fixed location (dataset-1) and non-fixed location (dataset-2). In detail, we asked different numbers of volunteers to stand once at each of the fixed 60 locations and collected CSI data lasting 15 s, and then added corresponding labels to the data for the category of each number of volunteers. Thus, we obtained 540 samples (9 Classes × 60 locations). For the second batch of data, we loosened the restriction by permitting volunteers to stand wherever within the CSI data collection area. In a similar number, we collected 540 samples and assigned each sample to the corresponding class of people. Specifically, the camera remained active during the whole data gathering process.

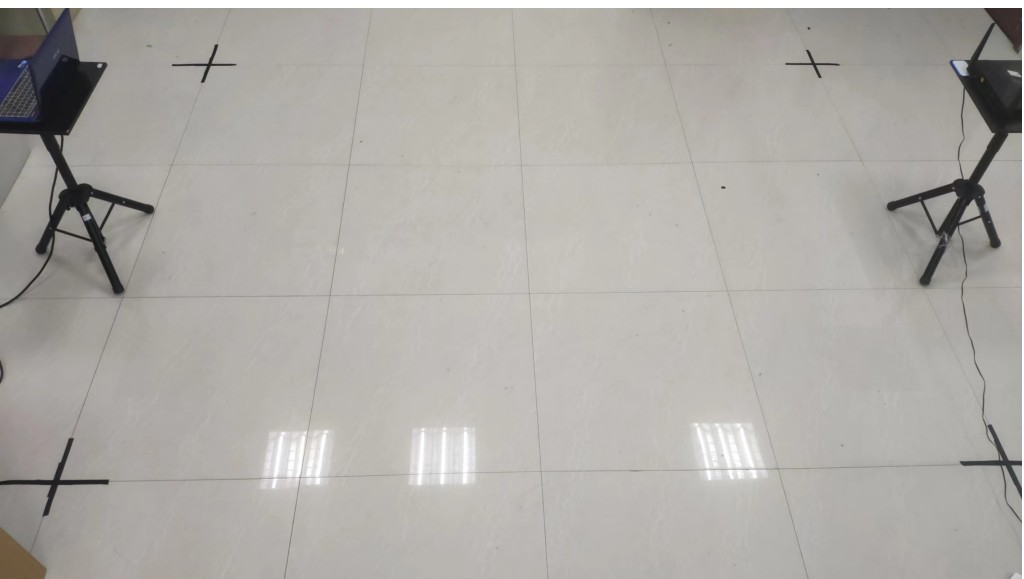

**Figure 2.** The area where the CSI data were collected. The crossbelt marks the boundary of the area, which is 2.4 m × 3.2 m in size.

*4.2. Preprocessing Details of the Dataset and CM-NET Hyperparameter Settings*

We extract the CSI amplitude data from the CSI dataset and preprocess it with noise and outlier removal. After that, we segmented the CSI dataset based on a 3 s time window to expand the dataset. Since the location and number of people do not change in each video sample we collect, so we sample fixed interval frames from the original video data as our video dataset. Specifically, we conduct sampling every 3 s so that each CSI sample has a frame to correspond to. This provides the benefit of speeding up CM-NET training while conserving computational resources.

The teacher and student networks are trained as part of the CM-NET training procedure. We first use the COCO dataset to establish the initial training weights for the teacher network, and then we use the video dataset we collected to fine-tune them. The weight parameters of the teacher network are frozen once the training process is complete. Before feeding the CSI data into CM-NET synchronously with the corresponding video frames, we load the teacher network's weight parameters into it to train the student network. In the course of training the student network, the soft labels generated by the teacher network and the tagged real labels are used in conjunction to supervise the training process.

CM-NET training is performed on a laptop with an RTX 2060 GPU, based on the PyTorch deep learning framework. The batch size for training the student network is set to 8, the training parameters are optimized using the Adam optimizer, the learning rate is set to 0.001, the MCAT is set to 5 layers, and the balance factor $\lambda$ is set to 0.5. If not stated otherwise, the following results are assessed using a 7:1:2 random allocation for the training, testing, and validating datasets.

Performance of CM-NET under Different Parameters

(1) The impact of different $\lambda$. The magnitude of $\lambda$ determines how far the teacher network can affect the student network during training. In this paper, we selected four different degrees and tested them on dataset-1, and the results are shown in Figure 3a. The performance of CM-NET with five degrees is lighter (0.3): 0.8550, light (0.4): 0.8515, moderate (0.5): 0.8640, heavy (0.6): 0.8493, and heavier (0.7): 0.6374. 0.6374. Not surprisingly, the performance of the student network did not rise as the effect of the teacher network increased. Only appropriate instruction improves the ability of the student network to extract the characteristics of the number of people. Suppose the teacher network interferes too much with the learning of the student network. In that case, it will make the students depend too much on the teacher network's inference when extracting features, which leads to the student network not paying enough attention to its data, and the model's performance decreases.

(2) The impact of MCAT with different layers. The number of layers of MCAT determines the depth of the CM-NET network. In this paper, we selected four different depths and performed tests on dataset-1, whose results are shown in Figure 3b. The performance of CM-NET with four depths is four layers: 0.8540, five layers: 0.8640, six layers: 0.8654, and seven layers: 0.8512. Obviously, the depth of the network does affect the ability of CM-NET to extract features. For the same number of samples, an appropriate increase in the number of MCAT layers can boost the extraction ability of the network. However, increasing the number of MCAT layers without expanding the data can cause the network to overfit during the training process, which leads to worse prediction results. Notably, the final number of MCAT layers selected in this paper is five. This is because the accuracy of the MCAT of five layers is similar to that of six. However, five layers MCAT causes less training time, fewer parameters, and fits with less difficulty.

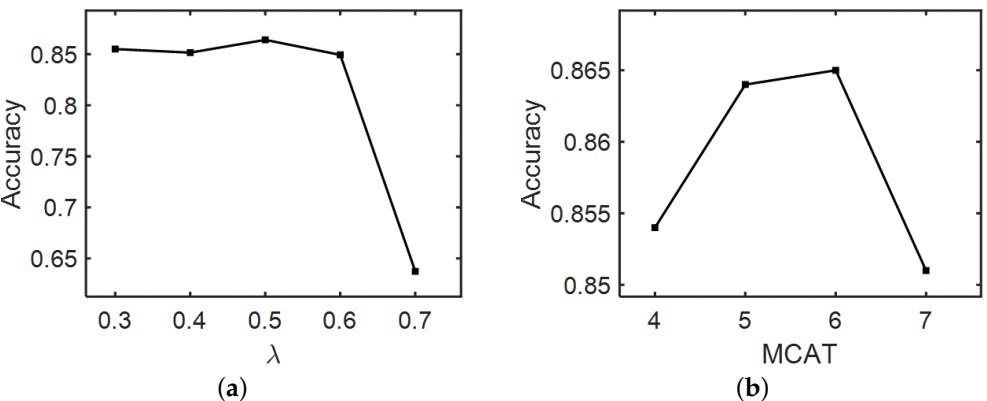

(a)      (b)

**Figure 3.** Performance of CM-NET under different parameters. (**a**) The impact of different $\lambda$. (**b**) The impact of MCAT with different layers.

*4.3. Experimental Results*

4.3.1. Performance Evaluation Metrics

In this paper, four evaluation metrics of Accuracy, Recall, Precision, and F1score are used to evaluate the people counting performance of CM-NET, and the expressions of each evaluation metric are as follows:

$$Accuracy = \frac{TP}{FP + FN + TP + TN} \tag{10}$$

$$Recall = \frac{TP}{TP + FN} \tag{11}$$

$$Precision = \frac{TP}{TP + FP} \tag{12}$$

$$F1\text{-}score = 2\frac{Precision \times Recall}{Precision + Recall} \tag{13}$$

where $TP$ represents the number of samples that were correctly classified by the classifier and rated positive by the classifier; $TN$ represents the number of samples that were correctly classified by the classifier and rated negative by the classifier; $FP$ represents the number of samples that were incorrectly classified by the classifier and rated positive by the classifier; and $FN$ represents the number of samples that were incorrectly classified by the classifier and rated negative by the classifier.

### 4.3.2. CM-NET Performance Evaluation

To test the performance of CM-NET at multiple locations, we use CM-NET and CM-NET (Training without image) on dataset-1 (shown in Table 1) and dataset-2(shown in Table 2), respectively.

**Table 1.** CM-NET performance on dataset-1.

| Model | Accuracy | Recall | Precision | F1-Score |
|---|---|---|---|---|
| CM-NET | 86.40% | 85.44% | 85.68% | 85.15% |
| CM-NET (Training without image) | 82.44% | 81.19% | 81.49% | 80.72% |

Table 1 compares the performance of CM-NET and CM-NET (Training without image) on dataset-1. On dataset-1, it is simple to observe that CM-NET (Training without image)'s people counting accuracy can only reach 82.44% and other metrics are around 81%. In contrast, CM-NET's accuracy can reach 86.40%, and other metrics are around 85%, which is superior to CM-NET (Training without image). By examining the test data, it can be seen that even if the position is fixed, the CSI amplitude is still subject to position interference. The CM-NET (Training without image) performs poorly in classification because it cannot extract more effective crowd features. In contrast, CM-NET was trained by introducing soft labels produced by the vision-based network. This allowed the network to extract a greater variety of people features, and its performance in counting was greatly enhanced.

**Table 2.** CM-NET performance on dataset-2.

| Model | Accuracy | Recall | Precision | F1-Score |
|---|---|---|---|---|
| CM-NET | 83.61% | 85.40% | 82.92% | 82.65% |
| CM-NET (Training without image) | 75.92% | 75.64% | 75.80% | 75.43% |

Table 2 compares the performance of CM-NET and CM-NET (Training without image) on dataset-2. We can see that on dataset-2, CM-NET (Training without image)'s accuracy in counting people can only reach 75.92%, and all other metrics are around 75%, while CM-NET's accuracy in counting people can reach 83.61%, and all other metrics are around 82%, which is superior to CM-NET (Training without image). The results shown in Table 2 indicate that location change interferes more with CSI when the location of the detected people is not fixed. As the location of detected people changes randomly, and we cannot collect data from all locations, the CM-NET (Training without image) network extracts information with significantly less attention being paid to the feature of the number of people, and its performance for counting people declines. While, the location change almost did not affect the performance of the vision-based network. CM-NET transfers the knowledge learned by the vision-based people counting network to the CSI-based network using the technique of knowledge distillation. By doing so, CSI-based networks can extract features more efficiently and count people more effectively.

We present the confusion matrix of CM-NET and CM-NET (Training without image) test results in Figure 3. Where Figure 4a shows the confusion matrix of CM-NET (Training without image) on dataset-1, Figure 4b the confusion matrix of CM-NET on dataset-1, Figure 4c the confusion matrix of CM-NET (Training without image) on dataset-1, and Figure 4d the confusion matrix of CM-NET on dataset-2.

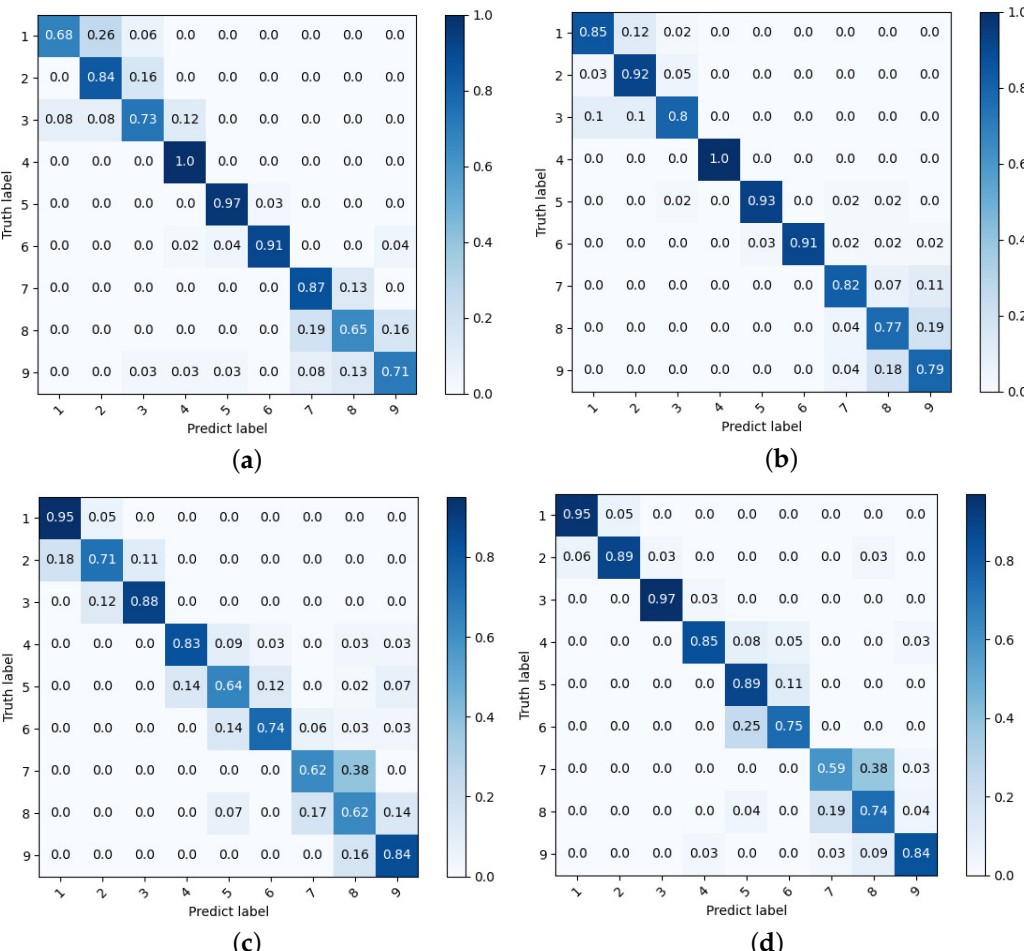

**Figure 4.** Confusion matrix of CM-NET (Training without image) and CM-NET test results. In the confusion matrix, the row coordinates represent the number of people predicted by the network and the vertical coordinates represent the actual number of people; the numbers in the matrix represent the proportion of correctly predicted samples to the total samples in the class. (**a**) CM-NET (Training without image) confusion matrix on dataset-1. (**b**) CM-NET confusion matrix on dataset-1. (**c**) CM-NET (Training without image) confusion matrix on dataset-2. (**d**) CM-NET confusion matrix on dataset-2.

It is clear from the confusion matrix in Figure 4 that CM-NET increases the CSI-based network's accuracy for classifying several people-counting groups. The improved accuracy is attributed to the fact that CM-NET makes the CSI-based network pay more attention to the features of the number of people when extracting features by introducing soft labels of visual information as supervisory signals, which attenuates the effect of changes in location on the network feature extraction.

### 4.3.3. Comparative Experiments and Correlation Analysis with Existing Related Work

We compared with some existing networks, including DNN [27], SVM, KNN, and the current state-of-the-art classification network Two-Stream Transformer [24], to show the superiority of CM-NET in people counting at multiple places.

The performance of the five current networks on dataset-1 is shown in Table 3. By analyzing Table 3, We found that on dataset-1, differences in location interfered with CSI, resulting all network trained using CSI only that could not achieve the desired recognition results. In contrast, CM-NET uses soft labels generated by the vision-based network to supervise the training of the CSI network, allowing the CSI network to efficiently extract more useful human feature data and the network to achieve produce more accurate results.

**Table 3.** Performance of existing networks on dataset-1.

| Model | Accuracy | Recall | Precision | F1-Score |
|---|---|---|---|---|
| CM-NET | 86.40% | 85.44% | 85.68% | 85.15% |
| Two-Stream Transformer [24] | 72.24% | 70.92% | 64.77% | 66.50% |
| DNN [27] | 53.88% | 33.33% | 17.96% | 23.34% |
| SVM [26] | 49.41% | 41.71% | 42.25% | 40.17% |
| KNN [26] | 49.55% | 41.83% | 41.45% | 41.37% |

The five current networks' performance tests are shown in Table 4 for dataset-2. Table 4, through analysis, revealed that on dataset-2, the interference caused by the location is more apparent due to the position's immobility, causing all networks' accuracy to fall. Each network metric has a pronounced degradation when CSI is solely employed, especially for machine learning networks, where performance is approximately halved. In contrast, CM-NET increases the network's ability to extract features about the number of people by transferring the information learned from the vision network to the CSI network, reducing interference caused by location changes and ensuring stability for the network's performance.

**Table 4.** Performance of existing networks on dataset-2.

| Model | Accuracy | Recall | Precision | F1-Score |
|---|---|---|---|---|
| CM-NET | 83.61% | 83.40% | 82.92% | 82.65% |
| Two-Stream Transformer [24] | 53.51% | 53.11% | 37.00% | 42.65% |
| DNN [27] | 39.73% | 33.33% | 13.24% | 18.96% |
| SVM [26] | 25.92% | 24.32% | 30.31% | 22.28% |
| KNN [26] | 22.84% | 22.16% | 22.37% | 22.25% |

The results in Tables 3 and 4 reveals that in both cases—especially when the location is not fixed—CM-NET achieves superior people counting performance.

## 5. Conclusions

In this paper, we introduce multimodal knowledge distillation to the task of CSI-based people counting. We propose an end-to-end supervised cross-modal learning network, CM-NET. By transferring knowledge from a vision-based network to a CSI-based network using deep learning and distillation learning, CM-NET enhances the performance of CSI-based networks for people counting. Our data acquisition uses only a pair of receivers, transmitters, and a camera, and experiments are conducted in a real room. The experimental results show that the average recognition accuracy of CM-NET for up to 9 people is 86%, which is better than existing related methods. Our method is quite robust in complex indoor environments. For our future work, we plan to further improve the recognition accuracy of people counting.

**Author Contributions:** Conceptualization, Supervision, Project Administration, Writing—Original Draft, J.G.; Formal Analysis, Methodology, Software, Writing—Original Draft, X.G.; Visualization, Investigation, Z.L.; Writing—Review & Editing, Investigation, M.J.; Data Curation, Formal Analysis, J.W.; Visualization, Writing—Review & Editing, X.Y.; Conceptualization, Funding Acquisition, Resources, Supervision, Writing—Review & Editing, P.X. All authors have read and agreed to the published version of the manuscript.

**Funding:** This research was funded by the National Key Research and Development Program of China grant number 2018YFB1802401.

**Institutional Review Board Statement:** Not applicable.

**Informed Consent Statement:** Informed consent was obtained from all subjects involved in the study.

**Data Availability Statement:** The data that support the findings of this study are available on request from the corresponding author. The data are not publicly available due to privacy or ethical restrictions.

**Acknowledgments:** This work was partially supported by National Natural Science Foundation of China under grant agreements Nos. 61902316, 62133012, 61973250, 62073218, 61973249, 61902313, 62002271. Shaanxi Provincial Department of Education serves local scientific research under 19JC038, the Key Research and Development Program of Shaanxi under 2021GY-077, the popular development of deep integration of digital economy industry and green ecological industry: 150012100001, Young science and technology nova of Shaanxi Province: 2022KJXX-73 and the Fundamental Research Funds for the Central Universities under grant No. XJS210310.

**Conflicts of Interest:** The authors declare no conflict of interest.

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
