# Peer review of "CM-NET: Cross-Modal Learning Network for CSI-Based Indoor People Counting in Internet of Things"

_electronics, doi:10.3390/electronics11244113_

Round 1

Reviewer 1 Report

Minor English language proofing for mostly typos would be beneficial to the work's legibility (just a few examples: almost all the possessives within the work require correction, "counting methods RSS based" -> "counting methods, RSS based", "dataset.Specifically,", "that Even if" and few others "here and there")

The 3rd par of Section 1 appears twice

The notions of teacher and student networks require, prior or with with their initial mention, some explanation.

Sentences like "A recently proposed solution [...] and phase information." require citations in order to stand.

To the untrained eye, Section 1's first few paragraphs would be better off by making more clear that the proposed method does not rely on counted people having with them a wifi enabled device, and thus the proposed method is not counting people by counting the connections of users' devices to some central node.

The experimentation would benefit from adding a subsection detailing quantitatively and qualitatively the effect of the parameters discussed in the last par of Section 4.2

Author Response

Response to Reviewer 1 Comments

Point 1: Minor English language proofing for mostly typos would be beneficial to the work's legibility (just a few examples: almost all the possessives within the work require correction, "counting methods RSS based" -> "counting methods, RSS based", "dataset.Specifically,", "that Even if" and few others "here and there")

Response 1: Thank you for the reminder.

Line 217, page 5 "CM-network NET's" -> " CM-NET 's ".

Line 176, page 4 "their" -> " they ".

Line 160, page 4 "counting methods RSS based" -> "counting methods, RSS based".

Line 278, page 8 "dataset.Specifically" -> " dataset. Specifically"

Line 340, page 10 "that Even if"-> " that even if "

Line 235, page 6 "here," -> " where,"

Line 371, page 11 " the numbers in the matrix represent how many people were correctly predicted to how many people there were." ->"the actual number of people; the numbers in the matrix represent
the proportion of correctly predicted sample."

Point 2: The 3rd par of Section 1 appears twice

Response 2: Thank you for the reminder. We removed the duplicate paragraph

Point 3: The notions of teacher and student networks require, prior or with with their initial mention, some explanation.

Response 3: Thank you for the underlinning this deficiency. We explain these two notions in line 57 ,page 2.

Point 4: Sentences like "A recently proposed solution [...] and phase information." require citations in order to stand.

Response 4: Thank you for the reminder. I have added references as per your comments.

line 51, page 2 “A recently proposed solution [21] and phase information.”

Point 5: To the untrained eye, Section 1's first few paragraphs would be better off by making more clear that the proposed method does not rely on counted people having with them a wifi enabled device, and thus the proposed method is not counting people by counting the connections of users' devices to some central node.

Response 5: Thank you for the reminder. To avoid any misunderstanding of the reader,we explain the principle of the WIFI-based approach in line 38, page 2.

Point 6: The experimentation would benefit from adding a subsection detailing quantitatively and qualitatively the effect of the parameters discussed in the last par of Section 4.2.

Response 6: Thank you for your suggestion. We followed your suggestion and added a subsection to section 4.2. In subsection, we have selected the most important parameters for the network to compare and analyze. This is reflected in line 295, page 9.

Reviewer 2 Report

Channel states and RSS are addressed on Predictive estimation of optimal signal strength from drones over IoT frameworks in smart cities, and An efficient channel reservation technique for improved QoS for mobile communication deployment using high altitude platform, i suggest authors discuss and compare with previous  

why you use teacher network and student network

in table 4, explain in text why the performence of proposed methods is the best 

Author Response

Response to Reviewer 2 Comments

Point 1: Channel states and RSS are addressed on Predictive estimation of optimal signal strength from drones over IoT frameworks in smart cities, and An efficient channel reservation technique for improved QoS for mobile communication deployment using high altitude platform, i suggest authors discuss and compare with previous 

Response 1: Thank you for your suggestion. We followed your suggestion for discussion and comparison with the previous. We discussion “Channel states and RSS are addressed on Predictive estimation of optimal signal strength from drones over IoT frameworks in smart citie” in line 180, page 4. We discussion “An efficient channel reservation technique for improved QoS for mobile communication deployment using high altitude platform” in line 200, page 5.

Point 2: why you use teacher network and student network

Response 2: Thank you for the underlinning this deficiency. We explained the problem in line 57 , page 2.

Point 3: in table 4, explain in text why the performence of proposed methods is the best.

Response 3: Thank you for the underlinning this deficiency. We explained the problem in line385, page 12.

Round 2

Reviewer 2 Report

Authors addressed my comments very well and it can be accept for publication